# Development of Hemispherical 3D Models of Human Brain and B Cell Lymphomas Using On-Chip Cell Dome System

**DOI:** 10.3390/bioengineering11121303

**Published:** 2024-12-23

**Authors:** Ryotaro Kazama, Rina Ishikawa, Shinji Sakai

**Affiliations:** Graduate School of Engineering Science, Osaka University, 1-3 Machikaneyama, Toyonaka 560-8531, Osaka, Japan; r.kazama@cheng.es.osaka-u.ac.jp (R.K.); rina.ishikawa@cheng.es.osaka-u.ac.jp (R.I.)

**Keywords:** lymphoma, Cell Dome, 3D cell cultures, tumor microenvironment, hypoxia, CD20

## Abstract

Lymphocytes are generally non-adherent. This makes it challenging to fabricate three-dimensional (3D) structures mimicking the three-dimensional lymphoma microenvironment in vivo. This study presents the fabrication of a hemispherical 3D lymphoma model using the on-chip Cell Dome system with a hemispherical cavity (1 mm in diameter and almost 300 µm in height). Both the human brain lymphoma cell line (TK) and human B cell lymphoma cell line (KML-1) proliferated and filled the cavities. Hypoxic regions were observed in the center of the hemispherical structures. CD19 expression did not change in either cell line, while CD20 expression was slightly upregulated in TK cells and downregulated in KML-1 cells cultured in the Cell Dome compared to those cultured in two-dimensional (2D) flasks. In addition, both TK and KML-1 cells in the hemispherical structures exhibited higher resistance to doxorubicin than those in 2D flasks. These results demonstrate the effectiveness of the on-chip Cell Dome for fabricating 3D lymphoma models and provide valuable insights into the study of lymphoma behavior and the development of new drugs for lymphoma treatment.

## 1. Introduction

Lymphoma is a type of blood cancer classified into two types: Hodgkin lymphoma and non-Hodgkin lymphoma. Non-Hodgkin lymphoma represents a wide range of diseases, ranging from mostly gradual to very aggressive malignancies, 80~90% of which are B cell-derived [1]. Two-dimensional (2D) models of lymphoma-derived B cells have been used to elucidate their cellular function in hematology/oncology and for drug screening in new drug development [2,3]. However, 2D models cannot mimic cellular interactions in the lymphoma microenvironment and may not provide a correct understanding of the cellular function of lymphoma-derived cells and the efficacy of compounds for medical treatment [2,4]. Thus, lymphoma models that better mimic the lymphoma microenvironment are needed to accurately elucidate the complex cellular functions of lymphoma-derived cells and to address challenges in lymphoma therapy.

Three-dimensional (3D) culture methods have gained more attention for studying tumor cell biology under conditions that closely resemble in vivo cell behavior compared to two-dimensional culture methods [5,6]. Three-dimensional culture methods such as hanging drops [7,8] and microwells [9] have been used mainly for solid tumor analysis, which mainly involves using adherent cells. However, lymphoma-derived cells are generally non-adherent cells, represented by hematopoietic cells and lymphocytes, and their weak cell adhesion properties make them difficult to apply to conventional 3D culture systems. This makes handling, culturing, and evaluating in 3D culture more complex. Therefore, it is challenging to fabricate 3D structures mimicking the 3D lymphoma microenvironment in vivo, and the field of 3D culture in lymphoma remains understudied, with only a few papers reported [2,10]. Although microwells [11], optical tweezers [12], and lymphoma-on-chip [13] allow for 3D culture of lymphoma-derived cells, they require gentle solution manipulation during culture, analysis, and evaluation due to the weak cell-to-cell adhesion of non-adherent cells. In addition, these methods also have difficulty producing 3D organized lymphoma-derived cells that are uniform in size. Therefore, there are no standardized and reproducible 3D lymphoma models that can be applied for a variety of applications, despite their potential to reveal resistance to lymphoma therapies and their mechanisms. Recently, we developed an on-chip cell culture and evaluation system called a “Cell Dome” which is fabricated on a glass substrate and features a dome with a hemispherical cavity (1 mm in diameter and almost 300 µm in height) that provides a 3D cell growth space for enclosed cells [14]. Some major advantages of the Cell Dome are that it can be applied to non-adherent cells cultured in suspension, enables 3D culture without relying on the adhesion ability of the cells themselves, and allows for easy solution handling without cell loss or damage, compared to conventional 3D culture methods such as hanging drops. In addition, multiple Cell Domes can be fabricated on the same glass plate, allowing for high-throughput analyses such as cell-based microarrays. We recently reported that the Cell Dome can be used as a cell-based transfection array with non-adherent cells suspended in a medium [15].

Based on the above, the aim of this study was to mimic the complex microenvironment of lymphoma tumors using 3D cell culture and to develop a novel platform for the study of lymphoma behavior and the development of new drugs for lymphoma treatment. This study presents the fabrication of a hemispherical 3D lymphoma model using the Cell Dome system with a hemispherical cavity (Figure 1). The Cell Dome offers distinct advantages over existing 3D culture techniques, including improved handling of non-adherent cells, ease of operation and reproducibility, and the ability to mimic the in vivo microenvironment more accurately. We used the human brain lymphoma cell line (TK cells) and the human B cell lymphoma cell line (KML-1 cells) as models to study cellular behavior in the Cell Domes, including cell proliferation, cellular hypoxia, and drug sensitivity. We also examined CD19 and CD20 expressions, biomarkers for immunotherapy targeting lymphomas that are mainly B cell-derived [16,17,18], on cells cultured in the Cell Domes. The Cell Dome is a promising tool for fabricating lymphoma models and hence for studying lymphoma behavior in 3D culture, providing valuable insights for drug development in lymphoma treatment.

## 2. Materials and Methods

### 2.1. Cell Culture

The human brain lymphoma cell line (TK cells: JCRB1206) and human B cell lymphoma cell line (KML-1 cells: JCRB1347) were obtained from the JCRB Cell Bank (Ibaraki, Japan). These cells were cultured in RPMI 1640 medium (Nissui, Tokyo, Japan) with the addition of 20% fetal bovine serum (Gibco, Grand Island, NY, USA) at 37 °C in a fully humidified atmosphere of 5% CO_2_.

### 2.2. Chemical Synthesis

The modification of Ph groups to sodium alginate (I-1G; mannuronic acid/guluronic acid ratio ≈ 0.7, at 1%; viscosity = 100–200 mPa·s, MW 70 kDa; Kimica, Tokyo, Japan) and gelatin from bovine skin (type B, approximately 250 g bloom; Sigma-Aldrich, St. Louis, MO, USA) was performed based on previous reports [19,20]. *N*-hydroxysuccinimide (Fujifilm Wako Chemical, Osaka, Japan) and water-soluble carbodiimide hydrochloride (Peptide Institute, Osaka, Japan) were used to synthesize alg-Ph by conjugating sodium alginate and tyramine hydrochloride (ChemImpex International, Wood Dale, IL, USA) and gela-Ph by conjugating gelatin and 3-(4-hydroxyphenyl) propionic acid (Tokyo Chemical Industry, Tokyo, Japan). The Ph contents of alg-Ph and gela-Ph were 2.7 × 10^−4^ and 7.8 × 10^−4^ mol-Ph/g, respectively.

### 2.3. Preparation of Cell Domes

Cell Domes were prepared based on previous reports [14]. Briefly, phosphate-buffered saline (PBS, pH 7.4, 1 µL) containing 50 U/mL horseradish peroxidase (HRP; 140 units/mg; Fujifilm Wako Chemical), 3.0 % *w*/*v* gelatin from porcine skin (type A, approximately 300 g bloom; Sigma-Aldrich), and 1.2 × 10^7^ cells/mL TK or KML-1 cells was spotted in annular water-repellent patterns (outer/inner diameter: 1.4/1.0 mm) printed on a glass plate. After forming this core gel at 5 °C for 15 min, PBS (10 µL) containing 0.5% *w*/*v* alg-Ph, 1.0% *w*/*v* gela-Ph, and 1 mM hydrogen peroxide (H_2_O_2_; Fujifilm Wako Chemical) was dropped on the core gels to form hydrogel shells through HRP-mediated hydrogelation. After leaving it to stand at 15 °C for 5 min, the Cell Domes formed on the glass plate were immersed in the medium containing 4000 U/mL catalase (from bovine liver; Fujifilm Wako Chemical). The medium was replaced every 2–3 days (Figure 1). FITC-labeled gela-Ph was used instead of gela-Ph for structural analysis of the Cell Domes. Images of TK and KML-1 cells cultured in the Cell Domes were captured at appropriate incubation periods using a fluorescence microscope (BZ 9000; Keyence, Tokyo, Japan). The number of cells in the Cell Domes was analyzed by degrading the hydrogel shells with 2.4 U/mL alginate lyase (Sigma-Aldrich) and collecting cells from the Cell Domes. They were analyzed using an automated cell counter (Countess II FL; Thermo Fisher Scientific, Waltham, MA, USA).

### 2.4. Histological Analysis

Cell Domes with cells incubated for 10 days were immersed in PBS containing 4.0% *w*/*v* paraformaldehyde (Fujifilm Wako Chemical) at 4 °C for 1 h. Then, they were immersed in PBS containing 5.0% *w*/*v* sucrose (Fujifilm Wako Chemical) at 4 °C for 1 h. After washing with PBS twice, they were immersed in the Tissue-Tek^®^ O.C.T. compound (Sakura Finetek Japan Co., Ltd., Tokyo, Japan) and immediately frozen in liquid nitrogen. Subsequently, the frozen specimens were cut into 15 µm thick sections using a cryotome (2800 Frigocut E; Reichert-Jung, Heidelberg, Germany). The sections were stained with hematoxylin and eosin and observed using a fluorescence microscope.

### 2.5. Initial Cell Density Analysis

PBS containing 50 U/mL horseradish peroxidase, 3.0% *w*/*v* gelatin from porcine skin, and 1.5 × 10^8^ cells/mL of TK or KML-1 cells was used as a core solution to fabricate the Cell Domes in the same manner as described in the “Preparation of Cell Domes” section. These Cell Domes, called “high-density Cell Domes”, were fabricated with an initial cell density of 1.5 × 10^8^ cells/mL. Images of TK and KML-1 cells cultured in the high-density Cell Domes were captured at appropriate incubation periods using the fluorescence microscope. The number of cells in the high-density Cell Domes was analyzed in the same manner as described in the “Preparation of Cell Domes” section. Additionally, the cells cultured for 3 days in the high-density Cell Domes were analyzed via hypoxia probe solution analysis, flow cytometry for CD19 and CD20 expressions, real-time PCR for the gene expression of hypoxia-inducible factor-1α (HIF-1α), and drug sensitivity analysis, as described below.

### 2.6. Hypoxia Probe Solution Analysis

Cell Domes with cells incubated for 10 days and high-density Cell Domes with cells incubated for 3 days were immersed in a medium containing 2 µM hypoxia probe solution (Medical & Biological Laboratories, Nagoya, Japan) for 24 h. After washing with PBS twice, they were observed using the fluorescence microscope. The 2D cultured cells stained with hypoxia probe solution were used as a control.

### 2.7. Flow Cytometry

The cells cultured for 2, 7, and 10 days in the Cell Domes and the cells cultured for 3 days in the high-density Cell Domes were collected as described in the “Preparation of Cell Domes” section. The collected cells were immersed in PBS (500 µL) with Hu Fc Block (5 µL, Becton, Dickinson and Company, Franklin Lakes, NJ, USA) at 4 °C for 30 min. Then, the cells were immersed in PBS (500 µL) containing anti-CD19 mouse-mono (HIB19) APC (5 µL, Gene Tex, Irvine, CA, USA), an anti-CD20 monoclonal antibody (2H7) FITC (5 µL, ThermoFisher, MA, USA), and 2 µM propidium iodide (Dojindo, Kumamoto, Japan) at 4 °C for 1 h. After washing with PBS twice, the cells were analyzed using flow cytometry (Accuri C6; BD Biosciences, Tokyo, Japan). The 2D cultured cells with and without the antibody treatment were used as a positive and negative control, respectively.

### 2.8. Real-Time PCR

The cells cultured for 10 days in the Cell Domes and the cells cultured for 3 days in the high-density Cell Domes were collected as described in the “Preparation of Cell Dome” section. The CellAmp Direct TB Green RT-qPCR Kit (Takara Bio, Shiga, Japan) was used to extract total RNA from the cells according to the manufacturer’s instructions. cDNA was synthesized from DNase-treated RNA using the PrimeScript™ FAST RT reagent Kit with gDNA Eraser (Takara Bio). The gene expression of HIF-1α was examined using real-time PCR, employing the TB Green^®^ Premix Ex Taq™ II FAST qPCR (Takara Bio) with that of 18S ribosomal RNA (18s rRNA) used as reference genes for normalization. For this purpose, the Thermal Cycler Dice^®^ Real Time System II (Takara Bio) was used. The forward and reverse primers, shown in Table 1, were obtained from Eurofins Genomics (Tokyo, Japan). The 2nd Derivative Maximum method was used to calculate the Ct value, and the ΔΔCt method was used for data analysis. As a control, 2D cultured cells in a flask were used.

### 2.9. Drug Sensitivity Analysis

Cell Domes with cells incubated for 10 days or high-density Cell Domes with cells incubated for 3 days were incubated for 72 h in the medium containing 10, 100, and 1000 nM doxorubicin (DOX; Fujifilm Wako Chemical). The cells were stained with Calcein-AM (Dojindo) and propidium iodide and collected as described in the “Preparation of Cell Domes” section. Cell viability was determined using the fluorescence microscope. The 2D cultured cells were used as a control.

### 2.10. Statistical Analysis

Statistical analyses of two or more data sets in this study were performed using Student’s *t*-test or one-way ANOVA with Tukey’s post hoc analysis. Values are expressed as mean ± standard deviation.

## 3. Results

### 3.1. Structure of Cell Dome

The height of the hemispherical cell growth cavity of the Cell Dome was 283 ± 19 µm and the thickness of the hydrogel shell was 104 ± 16 µm (Figure 2, *n* = 3).

### 3.2. Proliferation of Cells Enclosed in Cell Dome

Both TK and KML-1 cells proliferated in the Cell Domes throughout the incubation period (Figure 3a,b). After 10 days of culture, the cells filled the hemispherical cavities of the Cell Domes (Figure 3a–c). The proliferations were also confirmed by measuring the number of the enclosed cells (Figure 3d), which increased over time, reached a peak after 10 days of culture, and then remained nearly constant (Figure 3d).

Both TK and KML-1 cells also proliferated in the high-density Cell Domes as the incubation period increased (Figure 4a,b). In TK cells, after 3 days of culture, dark areas were observed throughout the inside of the high-density Cell Domes, indicating that cells were growing evenly and in 3D within the hemispherical cavities. In contrast, after 3 days of culture, as shown in Figure 4b, a dark area was observed in the center of the high-density Cell Domes. This indicates that the KML-1 cells proliferated within the center of the hemispherical cavities in 3D. The proliferations were also confirmed by measuring the number of enclosed cells, with the number of cells cultured in the high-density Cell Domes for 3 days being less than those cultured in the Cell Dome for 10 days and slightly more than those cultured in the Cell Dome for 7 days (Figure 3d and Figure 4c).

### 3.3. Hypoxia of Cells Enclosed in Cell Dome

Both TK and KML-1 cells in the Cell Domes stained with the hypoxia probe solution after 10 days of culture showed red fluorescence in the center of the Cell Dome, indicating hypoxia, whereas those in 2D flasks did not show red fluorescence (Figure 5a,b). The gene expression of HIF-1α was 12.7 ± 7.3-fold higher in TK cells and 9.1 ± 2.6-fold higher in KML-1 cells cultured in the Cell Domes for 10 days compared to those cultured in 2D flasks (Figure 5d, *n* ≥ 3, *p ** < 0.05). In TK cells cultured in the high-density Cell Domes for 3 days, the cells stained with hypoxia probe solution also showed red fluorescence throughout the inside of the high-density Cell Domes, indicating hypoxia. In KML-1 cells cultured in the high-density Cell Domes for 3 days, the cells stained with the hypoxia probe solution also showed red fluorescence in the center of the high-density Cell Domes, indicating hypoxia, the same behavior as that of the cells cultured in the Cell Dome for 10 days (Figure 5b,c). The gene expression of HIF-1α was 4.3 ± 0.7-fold higher in TK cells and 10.6 ± 6.8-fold higher in KML-1 cells cultured in the high-density Cell Domes for 3 days compared to those cultured in 2D flasks (Figure 5d, *n* ≥ 3).

### 3.4. CD19 and CD20 Expression on Cells Enclosed in Cell Dome

Appendix A present the mean fluorescence intensities of 2D cultured cells immunostained without anti-CD19 or anti-CD20 and with anti-CD19 or anti-CD20, and those of Cell Dome cultured and high-density Cell Dome cultured cells immunostained with anti-CD19 and anti-CD20. Flow cytometry revealed that CD19 expression on TK and KML-1 cells cultured for 2, 7, and 10 days in the Cell Domes was similar to that of those cultured in 2D flasks (Figure 6a and Appendix A, *n* = 3, *p* > 0.05). Additionally, flow cytometry revealed that CD19 expression on TK and KML-1 cells cultured for 3 days in the high-density Cell Domes was similar to that of those cultured in 2D flasks and in the Cell Domes for 10 days (Figure 6b and Appendix A, *n* = 3, *p* > 0.05).

The flow cytometry analysis of CD20 expression on TK cells cultured in the Cell Domes for 2, 7, and 10 days showed two peaks, indicating both higher expression and similar expression levels to those cultured in 2D flasks (Figure 6a and Appendix A). The lower peak increased with incubation period and was almost equal in percentage to the higher peak at 10 days of culture. The flow cytometry analysis of CD20 expression on TK cells cultured in the high-density Cell Domes for 3 days also showed two peaks, similar peaks to those cultured in Cell Domes for 2 and 7 days (Figure 6a and Appendix A). In contrast, CD20 expression on KML-1 cells cultured for 2, 7, and 10 days in the Cell Domes showed one peak and gradually decreased with increasing culture periods (Figure 6a and Appendix A, *n* = 3, *p* * < 0.05). CD20 expression on KML-1 cells cultured for 10 days in the Cell Domes showed lower expression levels than those cultured in 2D flasks (Figure 6a and Appendix A, *n* = 3, *p* * < 0.05). CD20 expression on KML-1 cells cultured in the high-density Cell Domes for 3 days also showed one peak, indicating lower expression levels than those cultured in 2D flasks, with slightly higher peaks than those cultured in the Cell Domes for 10 days (Figure 6b and Appendix A, *n* = 3).

### 3.5. Drug Treatment of Cells Enclosed in Cell Dome

Figure 7 shows the viability of cells exposed to 10, 100, and 1000 nM of DOX. Both TK and KML-1 cells cultured in the Cell Domes for 10 days and the high-density Cell Domes for 3 days showed significantly higher cell viability compared to those in 2D flasks (Figure 7a,b, *n* = 3, *p* * < 0.05). In particular, when exposed to 10 nM of DOX, the cells cultured in the Cell Domes for 10 days showed significantly higher cell viability than those in the high-density Cell Domes for 3 days (Figure 7a,b, *n* = 3, *p* * < 0.05).

## 4. Discussion

Two-dimensional models cannot mimic cellular interactions in the lymphoma microenvironment and may not provide a correct understanding of the cellular function of lymphoma-derived cells or the efficacy of compounds for medical treatment [2,4]. Thus, it is important to develop a 3D lymphoma model that better mimics the lymphoma microenvironment. Meanwhile, the 3D culture of lymphoma-derived cells, generally non-adherent cells, is complex, and the field of 3D culture in lymphoma is still understudied [2,10]. We previously reported an on-chip Cell Dome system that facilitates non-adherent cell culture and evaluation with ease [14]. This study aimed to develop a hemispherical 3D lymphoma model utilizing the Cell Dome system and to evaluate the lymphoma-derived cells.

We used alg-Ph and gela-Ph as the Cell Dome’s hydrogel shells. The alg-Ph and gela-Ph composite hydrogels exhibited high cell compatibility and good permeability to low-molecular-weight compounds [23,24], ensuring nutrient and oxygen supply to enclosed cells. In addition, the Cell Dome has a hemispherical structure, providing enclosed cells with a 3D cell growth space, the inside of which was filled with a medium (Figure 2). Both TK and KML-1 cells proliferated well in the Cell Domes, filling the hemispherical cavity after 10 days of culture, and maintained their shape (Figure 3). This result was consistent with our previous report on K562 cells, similar non-adherent cells, cultured in the Cell Dome [14]. Compared to conventional 3D culture methods like microwells [11] and optical tweezers [12], the Cell Dome system enables the preparation of uniform hemispherical cavities enclosing cells and facilitates essential operations, such as solution manipulation and cell collection. These findings reveal that the Cell Dome system is suitable for the 3D culture of lymphoma-derived non-adherent cells, providing highly reproducible, uniformly sized 3D cultured cells essential for evaluation. Furthermore, in this study, lymphoma-derived cells cultured in the Cell Dome were fixed and processed directly in the Cell Dome, and a histological section was prepared. In major hematologic tumor models, this analysis is difficult because the weak connections between cells are often disrupted.

Hypoxia is a typical feature of the tumor microenvironment [25] and is closely associated with drug resistance, increased aggressiveness, the promotion of metastatic potential, and tumor progression [25,26]. Tumor cells under hypoxic conditions overexpress HIF-1α [27], and not only solid tumors but also malignant lymphomas express high levels of HIF-1α, which is considered a poor prognostic factor [28,29]. Pangarsa et al. reported that hypoxia, as with many solid tumors, was present in diffuse large B cell lymphoma (a type of lymphoma) and emphasized the need for further research on its role as a potential pathogenic or prognostic marker in this type of blood cancer [30]. Figure 5 shows that the central cells in the Cell Dome were hypoxic, with an oxygen concentration gradient within the Cell Dome, and that the cells in the Cell Dome had higher HIF-1α gene expression than those cultured in 2D flasks. To our knowledge, this is the first report to reproduce a hypoxic condition and an oxygen concentration gradient in the 3D culture of lymphoma-derived cells. The Cell Dome system, which recapitulates hypoxia, may help elucidate the role of hypoxia in lymphomas.

CD19 and CD20 are specific to B cells in lymphoma and serve as biomarkers for immunotherapy-targeting B cell-derived diseases [16,18]. Therefore, investigating CD19 and CD20 expression on lymphoma-derived cells in 3D cultures can provide important knowledge for developing drugs targeting these biomarkers. Neither TK nor KML-1 cells cultured in the Cell Dome exhibited significant differences in CD19 expression levels compared to those cultured in 2D flasks (Figure 6a and Appendix A, *p* > 0.05). Muz et al. reported that CD19 expression on multiple myeloma cells differentiated from B cells was not affected by hypoxia [31]. In contrast, changes in CD20 expression levels during 3D culture differed depending on the cell type. Although the mechanisms underlying the differences in CD20 expression levels based on cell type have not been elucidated, the upregulation of CD20 expression on TK cells cultured in the Cell Dome could be due to hypoxia. Ahmed et al. reported that HIF-1α might regulate CD20 expression in lymphoma [32]. To further investigate the potential regulation of CD20 expression through hypoxia, future studies should include the use of hypoxia inhibitors or siRNA targeting HIF-1α to directly assess its role in modulating CD20 levels. Additionally, examining downstream signaling pathways activated under hypoxic conditions could provide deeper mechanistic insights into how hypoxia influences CD20 expression in different lymphoma cell types. Also, in this study, the two peaks present in the flow cytometry analysis of CD20 expression on TK cells cultured in the Cell Dome (Figure 6a and Appendix A) indicate both similar and higher expression levels compared to those cultured in 2D flasks. This suggests that hypoxic conditions in the Cell Dome upregulated CD20 expression on TK cells. Moreover, the proportion of Cell Dome cultured cells expressing CD20 at levels similar to those of 2D cultured cells increased with extended incubation periods (Figure 6). This downregulation of CD20 on TK cells could result from mechanical stress and changes in the integrin signaling network caused by 3D culture [33]. The downregulation of CD20 was also observed on KML-1 cells cultured in the Cell Dome, where CD20 expression was gradually downregulated with increasing culture periods. This is likely due to increased 3D cell–cell interactions [33]. Some B cell lymphoma patients are resistant to anti-CD20-targeting agents, including rituximab [34]. The mechanisms behind resistance to anti-CD20 agents have remained largely unknown but may involve the downregulation or loss of CD20 expression [35,36]. The results of this study may help elucidate CD20 regulation in lymphoma, as multiple mechanisms may be involved in CD20 regulation [37], and the Cell Dome will be beneficial for future experiments investigating this regulation, which is important for lymphoma treatment.

So far, this study has primarily considered cells cultured for 10 days in the Cell Dome. If the same behavior observed in cells cultured in the Cell Dome can be controlled in a shorter culture period, it would be possible to evaluate the lymphoma model more rapidly. Therefore, we attempted to shorten the culture period by fabricating a Cell Dome with a significantly increased initial cell density, which we called the high-density Cell Dome. The enclosed cells proliferated even under a high-cell-density condition, and the incubation time taken for the cells to grow in 3D in the hemispherical cavities of the Cell Dome was reduced (Figure 4). Additionally, there were no significant differences in hypoxia levels, HIF-1α gene expression, or CD19 expression between the cells cultured for 3 days in the high-density Cell Dome and those cultured for 10 days in the Cell Dome, despite differences in cell density (Figure 3, Figure 4, Figure 5 and Figure 6). These findings suggest that the cells can proliferate effectively in a 3D hemispherical cavity with a high initial cell density, even during shorter incubation periods, and that their behavior can be evaluated comparably to those cultured for 10 days in the Cell Dome. CD20 expression on cells cultured in the high-density Cell Dome was slightly higher than that in cells cultured in the Cell Dome for 10 days (Figure 6). This suggests that the short culture periods in the high-density Cell Dome contribute to the downregulation of CD20 expression and that the mechanical stress and changes in the integrin signaling network induced by 3D culture, as described above, were less than those after 10 days of culture in the Cell Dome [33]. The above studies suggest that by significantly increasing the initial cell density in the Cell Dome preparation process, cells exhibiting almost identical behavior to Cell Dome cultured cells can be harvested in short periods. This would enable the rapid evaluation of lymphoma models utilizing the Cell Dome system. Another method of collecting cells that behave similarly to Cell Dome cultured cells in a short period involves miniaturizing the Cell Dome by mechanizing its preparation process and changing the size of the water-repellent ring pattern printed on the glass plates.

DOX is an effective lymphoma drug; it has been evaluated and widely used in combination therapy for lymphoma malignancies [38,39]. Therefore, we investigated the DOX sensitivity of cells cultured in the Cell Domes. The hydrogel shell of the Cell Dome has good permeability for low-molecular-weight compounds [23,24], so DOX can be delivered to the enclosed cells simply by adding it to the medium surrounding the Cell Dome. The cells exposed to DOX in the Cell Domes and high-density Cell Domes showed higher cell viability than those in 2D flasks (Figure 7, *n* = 3, *p* * < 0.05), suggesting that they acquired drug resistance due to the hypoxic conditions caused by the 3D culture of lymphoma-derived cells. Hypoxia and HIF-1α upregulated under hypoxic conditions promote tumor cell survival and chemotherapy resistance in many malignancies [40,41]. Tumor hypoxia and the upregulation of HIF-1α gene expression upregulate the expression of multidrug-resistant proteins and have been suggested to be the main cause of multidrug resistance [40,41,42]. Further research into the mechanisms influencing drug resistance acquisition, including on multidrug-resistant proteins, could provide deeper insights into the subject through lymphoma-derived cells cultured in the Cell Dome. Since the 2D culture system cannot effectively replicate the environment that induces multidrug resistance [43,44], drug resistance to DOX in cells cultured in the Cell Domes reflects the characteristics of the in vivo lymphoma microenvironment. In addition, the Cell Dome’s high stability and cytocompatibility allow it to be used for long-term culture and evaluation [14,24]. This suggests that this 3D lymphoma model utilizing the Cell Dome would be useful as a platform for the development of new drugs for lymphoma treatment.

The limitations of this study are as follows: The Cell Dome was fabricated manually and only lymphoma-derived cells were used. Our research group is currently investigating the mechanization of the Cell Dome fabrication process using an inkjet printer to achieve miniaturization, high throughput, and higher reproducibility. Furthermore, in the tumor microenvironment, including that of lymphoma, different cell types interact through crosstalk [11,33,44]. Therefore, 3D co-culturing of lymphoma-derived cells with other types of cells would better replicate the lymphoma microenvironment. We are currently investigating a 3D co-culturing lymphoma model with other cell types using the Cell Dome system.

## 5. Conclusions

In conclusion, a 3D lymphoma model was developed by culturing lymphoma-derived TK and KML-1 cells using the on-chip Cell Dome system. The central cells in the Cell Dome were hypoxic, with an oxygen concentration gradient within the Cell Dome. To our knowledge, this is the first study to reproduce an oxygen concentration gradient in 3D culture of lymphoma-derived cells that more faithfully recapitulates the lymphoma environment compared to conventional 2D culture methods. CD19 expression on either cell line cultured in the Cell Dome did not change compared to those cultured in 2D flasks. CD20 expression was slightly upregulated in TK cells and downregulated in KML-1 cells cultured in the Cell Dome compared to those cultured in 2D flasks. In addition, both TK and KML-1 cells cultured in the Cell Dome exhibited higher resistance to DOX compared to those cultured in 2D flasks. These results are attributed to the oxygen concentration gradient and 3D cell–cell interaction resulting from the 3D culture of lymphoma-derived cells, which are non-adherent cells, using the Cell Dome which cannot be reproduced using conventional 2D culture methods. This study demonstrates the utility of the Cell Dome in a 3D lymphoma model and provides valuable insights for studying lymphoma behavior and developing new drugs for lymphoma treatment. Novel 3D lymphoma models generated using the Cell Dome system will facilitate research into signal transduction, CD20 regulatory mechanisms, and the production of 3D co-culture models, significantly contributing to the comprehensive analysis of the biological functions of lymphoma-derived cells and the development of lymphoma research and therapies.

## Figures and Tables

**Figure 1 bioengineering-11-01303-f001:**
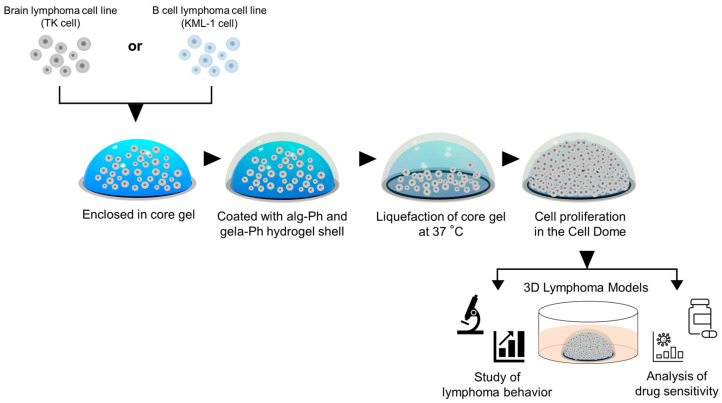
A schematic illustration of the Cell Dome preparation process. The process includes the preparation of the core gel with horseradish peroxidase (HRP), followed by the formation of the hydrogel shell with alg-Ph and gela-Ph through HRP-mediated hydrogelation, and finally, the immersion in the medium containing catalase for cell culture.

**Figure 2 bioengineering-11-01303-f002:**
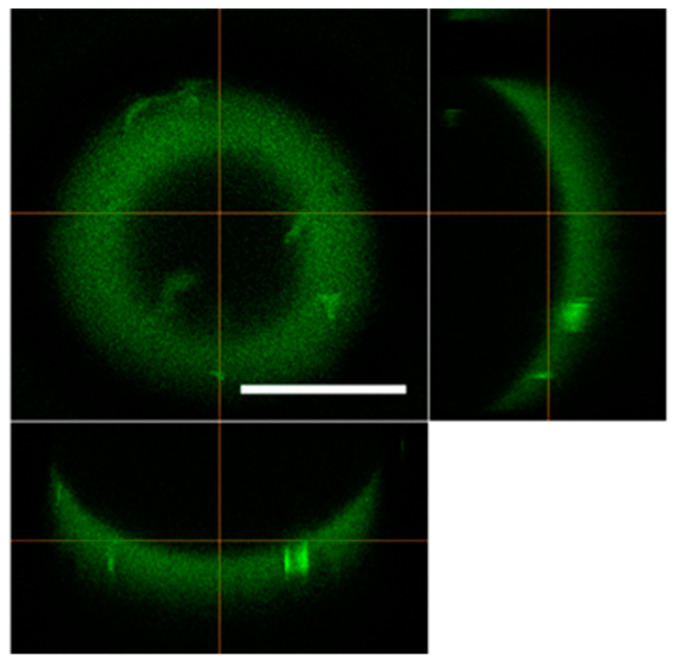
A confocal microscope image of the Cell Dome fabricated using a mixture of alg-Ph and gela-Ph as the hydrogel shell material. The image shows the uniform hemispherical structure of the Cell Dome. The bar in the panel represents 500 µm.

**Figure 3 bioengineering-11-01303-f003:**
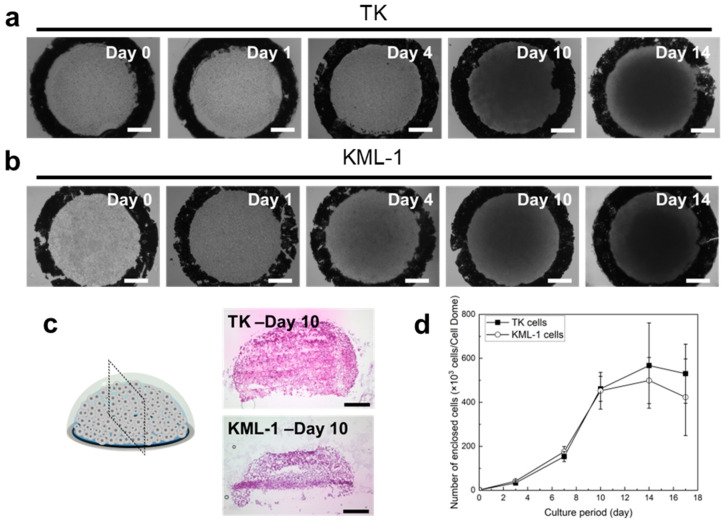
Microscopic images of TK (**a**) and KML-1 (**b**) cells cultured in the Cell Domes. (**c**) Histological section images of TK and KML-1 cells cultured in the Cell Domes for 10 days, showing the cell distribution within the cavities. (**d**) The absorbance values attributed to the mitochondrial activities of TK and KML-1 cells in the Cell Domes. The bars in panels (**a**–**c**) represent 250 µm. The bars in panel d represent the standard deviation (*n* = 3).

**Figure 4 bioengineering-11-01303-f004:**
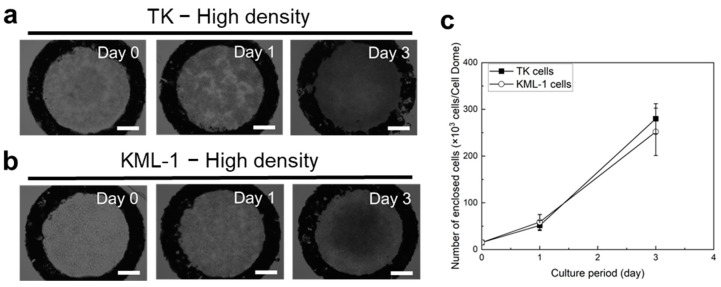
Microscopic images of TK (**a**) and KML-1 (**b**) cells cultured in the high-density Cell Domes. (**c**) The absorbance values attributed to the mitochondrial activities of TK and KML-1 cells in the high-density Cell Domes. The bars in panels (**a**,**b**) represent 250 µm. The bars in panel (**c**) represent the standard deviation (*n* = 3).

**Figure 5 bioengineering-11-01303-f005:**
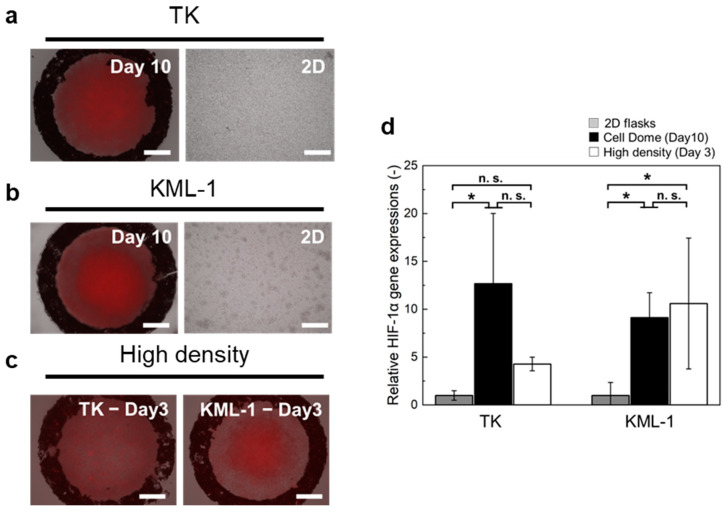
Fluorescence microscope images of TK (**a**) or KML-1 (**b**) cells cultured in the Cell Dome for 10 days and in a 2D flask, stained with hypoxia probe solutions. Red fluorescence indicates hypoxic conditions. (**c**) Fluorescence microscope images of TK or KML-1 cells cultured in the high-density Cell Domes for 3 days stained with hypoxia probe solutions. Red fluorescence indicates hypoxic conditions. (**d**) The relative gene expression of HIF-1α in TK or KMl-1 cells cultured for 10 days in the Cell Dome and cultured for 3 days in the high-density Cell Domes. The bars in panels (**a**–**c**) represent 250 µm. The bars in panel (**d**) represent the standard deviation (*n* ≥ 3, * *p* < 0.05, n.s.: *p* > 0.05).

**Figure 6 bioengineering-11-01303-f006:**
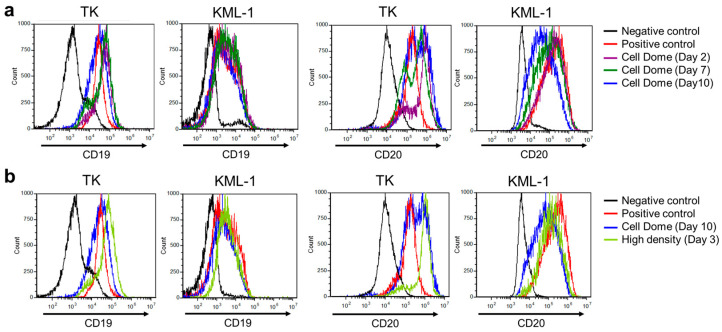
Flow cytometry analysis of CD19 and CD20 expression on 2D cultured TK or KMl-1 cells immunostained without anti-CD19 or anti-CD20 (negative control) and with anti-CD19 or anti-CD20 (positive control). (**a**) Cell Dome cultured cells for 2, 7, and 10 days immunostained with anti-CD19 or anti-CD20, and (**b**) high-density Cell Dome cultured cells for 3 days immunostained with anti-CD19 or anti-CD20.

**Figure 7 bioengineering-11-01303-f007:**
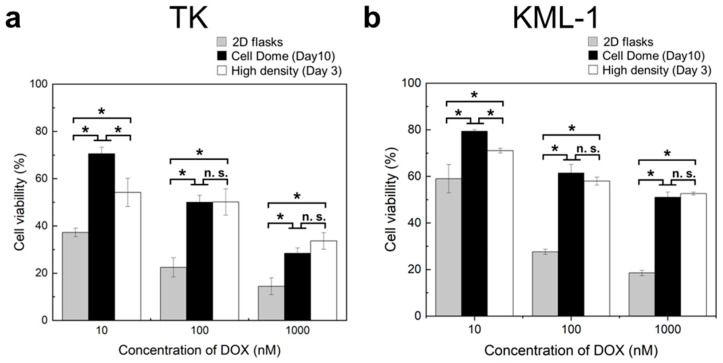
Viability of TK (**a**) or KML-1 (**b**) cells cultured in Cell Domes for 10 days, in high-density Cell Domes for 3 days, and in 2D flasks, with exposure to 10, 100, and 1000 nM doxorubicin (DOX). Bars represent standard deviation *(n* = 3, * *p* < 0.05, n.s.: *p* > 0.05).

**Table 1 bioengineering-11-01303-t001:** Primer used for detecting gene expressions.

Gene	Forward Primer	Reverse Primer	Reference
*18s rRNA*	5′-CCC GAC CCG GGG AGG TAG TG-3′	5′-GCC GGG TGA GGT TTC CCG TG-3′	[21]
*HIF-1α*	5′-TGC ATC TCC ATC TCC TAC CC-3′	5′-CCT TTT CCT GCT CTG TTT GG-3′	[22]

## Data Availability

The raw data supporting the conclusions of this article will be made available by the authors on request.

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
