# Peer review of "Development of Hemispherical 3D Models of Human Brain and B Cell Lymphomas Using On-Chip Cell Dome System"

_bioengineering, 2024, doi:10.3390/bioengineering11121303_

Round 1
Reviewer 1 Report
Comments and Suggestions for Authors
This study demonstrates the fabrication of a hemispherical 3D lymphoma model using an on-chip Cell Dome system. Both TK and KML-1 cells proliferated in the cavity, showing hypoxic regions. CD19 expression remained unchanged, while CD20 expression varied. Cells in the 3D model displayed higher doxorubicin resistance compared to 2D culture. The on-chip Cell Dome proves effective for creating 3D lymphoma models, offering insights into lymphoma behavior and drug development. Before considering this manuscript for publication, the authors should consider the following points in any revision as follows:
1. In reference 21, there was no description of the preparation process for cell domes; instead, it referenced another paper. Please note that when citing references, it is generally expected to cite the information from the original paper.
2. Is the area of the water-repellent part controllable? What range of sizes can be prepared?
3. In Figures 3a and 3b, the brightly colored outer ring of the circular pattern, what material is it made of? Why are there cracks appearing in some of the rings? For example, in Figure 3b, cracks appeared on the first day but reduced by the 14th day.
4. Despite using hydrogel materials, how can the survival rate of internal cells be ensured in 3D cell cultures? And is there any difference in drug permeability
Comments on the Quality of English LanguageNo
Author Response
Dear Reviewer,
Thank you for taking the time to review our manuscript and for your valuable feedback. We have modified our manuscript based on the comments. The following are detailed responses to each comment.
Response to Reviewer 1
This study demonstrates the fabrication of a hemispherical 3D lymphoma model using an on-chip Cell Dome system. Both TK and KML-1 cells proliferated in the cavity, showing hypoxic regions. CD19 expression remained unchanged, while CD20 expression varied. Cells in the 3D model displayed higher doxorubicin resistance compared to 2D culture. The on-chip Cell Dome proves effective for creating 3D lymphoma models, offering insights into lymphoma behavior and drug development. Before considering this manuscript for publication, the authors should consider the following points in any revision as follows:
Comment 1) In reference 21, there was no description of the preparation process for cell domes; instead, it referenced another paper. Please note that when citing references, it is generally expected to cite the information from the original paper.
Reply) Thank you for pointing out our mistake. We sincerely apologize for our mistake. As you pointed out, we have changed to cite information from the original paper regarding references in the “Preparation of Cell Dome” section (Page 3, Line 98).
Comment 2) Is the area of the water-repellent part controllable? What range of sizes can be prepared?
Reply) Thank you for your valuable comments. The area of the water-repellent ring pattern printed on the glass plate can be controlled. The size can be larger or smaller than the current size (outer/inner diameter: 1.4/1.0 mm). When reducing the size of the water-repellent ring pattern, mechanization of the Cell Dome fabrication process should also be considered as it would be difficult to fabricate a smaller Cell Dome manually. Based on your comments, we have added text to the “Discussion” section on these topics (Page 10, Line 393-396).
Comment 3) In Figures 3a and 3b, the brightly colored outer ring of the circular pattern, what material is it made of? Why are there cracks appearing in some of the rings? For example, in Figure 3b, cracks appeared on the first day but reduced by the 14th day.
Reply) Thank you for your valuable comments. The water-repellent ring pattern is printed on the glass plate, and the bright color outside the water-repellent ring pattern was an observation of the glass plate area where no Cell Dome was present. Cracks appearing in some of the rings were due to unintentional deterioration. However, the cracks do not affect Cell Dome preparation, structure, cell proliferation, and their analysis.
Comment 4) Despite using hydrogel materials, how can the survival rate of internal cells be ensured in 3D cell cultures? And is there any difference in drug permeability.
Reply) Thank you for your valuable comments. We apologize for the lack of explanation on our part. The Cell Dome has a hemispherical cavity filled with culture medium and covered with a hydrogel shell. The hydrogel shell has good permeability to oxygen, nutrients, and other low molecular weight compounds. Therefore, the hemispherical cavity in which cells are enclosed is a state in which non-adherent cells are suspended in a medium similar to a normal culture environment, thus ensuring a high cell survival rate. Due to the good permeability of the hydrogel, low molecular weight compounds such as doxorubicin (543.52 g/mol) used in this study are well permeable. Drug permeability depends on the molecular weight of the drug. Based on your comments, we have added a detailed description to the text (Page 9, Line 306-308, and Page 11, 399-402).

Reviewer 2 Report
Comments and Suggestions for Authors
The authors reported the fabrication of a hemispherical 3D lymphoma model using an on-chip cell dome system. The study demonstrated cell viability, protein expression and drug resistance.
1) In my opinion, the work simply reported the deployment of the cell dome system for TK and KML-1 cells. Therefore, regarding the technology, it is not new idea. It is similar to the hanging-drop as well. If there is a comparison with the hanging-drop (as the gold standard for 3D model), it will be clear about the benefit of the cell-dome system.
2) In the article, there are two models such as cell-dome (Fig.3) and high-density cell-dome (Fig.4). The explanation about the difference is unclear. The reason that we need to compare them should be described.
3) In Fig.3d, the uncertainty at the day after 10 days was very large. It suggested that the reproducibility of the test (for long run) is not good. In some cases, the cells keep growth, but they die in some cases. The authors should explain the reason that the other test (in Fig.4) is limited within 3 days.
4) The results showed that the deployment in cell-dome system is better than 2D model. The authors should explain about the reason behind in details (or hypothesis) to support the results.
5) In summary, there is no new idea. However, the work is systematically conducted. The results are useful for research community.
Author Response
Dear Reviewer,
Thank you for taking the time to review our manuscript and for your valuable feedback. We have modified our manuscript based on the comments. The following are detailed responses to each comment.
Response to Reviewer 2
The authors reported the fabrication of a hemispherical 3D lymphoma model using an on-chip cell dome system. The study demonstrated cell viability, protein expression and drug resistance.
Comment 1) In my opinion, the work simply reported the deployment of the cell dome system for TK and KML-1 cells. Therefore, regarding the technology, it is not new idea. It is similar to the hanging-drop as well. If there is a comparison with the hanging-drop (as the gold standard for 3D model), it will be clear about the benefit of the cell-dome system.
Reply) Thank you for your valuable comments and suggestions. Based on your suggestion, we have added a sentence to the “Introduction” section regarding a comparison with hanging drops (the gold standard for 3D models) to clarify the benefit of the Cell Dome system (Page 2, Line 56-59).
Comment 2) In the article, there are two models such as cell-dome (Fig.3) and high-density cell-dome (Fig.4). The explanation about the difference is unclear. The reason that we need to compare them should be described.
Reply) Thank you for your valuable comments and we apologize for not explaining clearly the difference between the two models. Based on your suggestion, we have added an explanation of the purpose of comparing these two models (Page 10, Line 371-376).
Comment 3) In Fig.3d, the uncertainty at the day after 10 days was very large. It suggested that the reproducibility of the test (for long run) is not good. In some cases, the cells keep growth, but they die in some cases. The authors should explain the reason that the other test (in Fig.4) is limited within 3 days.
Reply) Thank you for your valuable comments. In Figure 3d, the uncertainty on the day after 10 days would depend on the size of the hemispherical cavity of the Cell Dome (hemispherical cavity height; 283 ± 19 µm). Reducing the error rate of the hemispherical cavity height (about 0.07%) would reduce the uncertainty in Figure 3d. As you have pointed out, a limitation of this study is the uncertainty caused by the fact that Cell Dome is manufactured manually. To overcome this challenge, our research group is currently investigating the mechanization of Cell Dome manufacturing using an inkjet printer. We apologize for the lack of explanation of the limitations of this study and have added text to the article (Page 11, Line 420-424).
Regarding the long-term cultivation and evaluation, previous reports reported that cells cultured in the Cell Dome maintain their structure even after long-term (about 1 month) incubation, and this Cell Dome system can be used for long-term incubation [14, 24]. We apologize for the lack of explanation and have added text to the article based on your comments (Page 11, Line 415-417).
Figure 4 (High-density-Cell Dome) was considered within 3 days of culture, because it is a study designed to collect cells that behave like cells cultured in a Cell Dome in a shorter incubation period. We apologize for the lack of explanation and have added clarification of the purpose of the Figure 4 (High-density-Cell Dome) consideration in the text (Page 10, Line 371-375).
Comment 4) The results showed that the deployment in cell-dome system is better than 2D model. The authors should explain about the reason behind in details (or hypothesis) to support the results.
Reply) Thank you for your valuable comments and suggestions. Based on your comments, we have added the statement in the “Conclusion” section to explain the reason for supporting the results (Page 12, Line 433-435, and 440-443).
In summary, there is no new idea. However, the work is systematically conducted. The results are useful for research community.

Round 2
Reviewer 1 Report
Comments and Suggestions for Authors
The author has provided a detailed response to the previous feedback and is able to accept it
Reviewer 2 Report
Comments and Suggestions for Authors
I think the quality of the manuscript is acceptable for publication.